# Developmental fine-tuning of medial superior olive neurons mitigates their predisposition to contralateral sound sources

**Martijn C. Sierksma, J. Gerard G. Borst** *

Department of Neuroscience, Erasmus MC, Erasmus University Medical Center, Rotterdam, the Netherlands

* g.borst@erasmusmc.nl

## Abstract

Having two ears enables us to localize sound sources by exploiting interaural time differences (ITDs) in sound arrival. Principal neurons of the medial superior olive (MSO) are sensitive to ITD, and each MSO neuron responds optimally to a best ITD (bITD). In many cells, especially those tuned to low sound frequencies, these bITDs correspond to ITDs for which the contralateral ear leads, and are often larger than the ecologically relevant range, defined by the ratio of the interaural distance and the speed of sound. Using in vivo recordings in gerbils, we found that shortly after hearing onset the bITDs were even more contralaterally leading than found in adult gerbils, and travel latencies for contralateral sound-evoked activity clearly exceeded those for ipsilateral sounds. During the following weeks, both these latencies and their interaural difference decreased. A computational model indicated that spike timing-dependent plasticity can underlie this fine-tuning. Our results suggest that MSO neurons start out with a strong predisposition toward contralateral sounds due to their longer neural travel latencies, but that, especially in high-frequency neurons, this predisposition is subsequently mitigated by differential developmental fine-tuning of the travel latencies.

**Data Availability Statement:** Code needed to generate Figs 4 and S3 and data underlying the other figures is available at https://doi.org/10.5281/zenodo.10729468.

## Introduction

We use the tiny difference in arrival time of sounds at both ears as an important cue for the localization of sounds. Sounds coming from straight ahead will arrive simultaneously, whereas sounds originating from one side will arrive first at the ear nearest to the sound source. The maximal interaural time difference (ITD) is a function of the speed of sound and the head size. For humans, it is approximately 0.75 ms; for a rodent such as the Mongolian gerbil, with their much smaller head, only approximately 0.13 ms [1]. This maximal difference thus defines the ecological ITD range for sound localization across different species.

Mongolian gerbils can reliably detect ITDs behaviorally for sound frequencies of 0.75 to 2 kHz [2–4]. The ITDs of low-frequency sounds are decoded by a specialized brainstem circuit, in which the principal neurons of the medial superior olive (MSO) play a central role. MSO neurons receive bilateral excitatory inputs from spherical bushy cells of the anteroventral

**Funding:** This work was supported by the Nederlandse Organisatie voor Wetenschappelijk Onderzoek (NWO; ALW-open, 'Great Timing', #824.15.008 to JGGB; ENW-XS, 'All ears', OCENW. XS22.151 to MCS and ENW-M, 'Learning to localize sounds', OCENW.M.21.218 to JGGB). The funders had no role in study design, data collection and analysis, decision to publish, or preparation of the manuscript.

**Competing interests:** The authors have declared that no competing interests exist.

**Abbreviations:** AVCN, anteroventral cochlear nucleus; bITD, best interaural time difference; EPSP, excitatory postsynaptic potential; FSL, first-spike latency; ITD, interaural time difference; LNTB, lateral nucleus of the trapezoid body; MNTB, medial nucleus of the trapezoid body; MSO, medial superior olive; STDP, spike timing-dependent plasticity.

cochlear nucleus (AVCN) and bilateral inhibitory inputs from the lateral nucleus of the trapezoid body (LNTB) and medial nucleus of the trapezoid body (MNTB; Fig 1A). MSO neurons have a spindle shape with dendrites oriented mediolaterally [5,6]. Excitatory inputs from both sides are segregated to different dendrites [7,8]. The principal MSO neuron functions as a coincidence detector by summating the fast, monaural excitatory postsynaptic potentials (EPSPs) originating from both ears; the summed potential determines its spiking output [9–11]. The timing of EPSPs reflects the ITD and changes the spiking output. The ITD at which the spiking output becomes maximal is called the neuron's best ITD (bITD).

The view on how MSO neurons encode sound source direction along the azimuth has long been dominated by Jeffress' hypothesis [12,13]. This coding scheme holds that each neuron is tuned to a specific sound source direction ("peak coding"). For sounds coming from this direction, the internal delay, which is the net difference between the total travel times from each ear, exactly compensates for the difference in arrival time of the sound at both ears. In the MSO, many neurons have a bITD that represents a contralateral sound source direction [14–19], meaning that the travel times from the contralateral ear exceed those from the ipsilateral ear. However, there are also neurons that do not match Jeffress' predictions. Especially in small animals and for cells tuned to low sound frequencies, the internal delay can be larger than any ITD in the ecological range [17,19–21]. This means that these cells will increase their firing as sounds are coming increasingly from the contralateral side. These observations have led to a coding scheme called "slope coding" [22]. In slope coding, MSO neurons encode how contralateral a sound source is, but maximum firing is never attained [10,14–16,18–24]. The mechanisms underlying the preference for contralateral sounds have been heavily debated, and axonal conduction [8,13,25], well-timed synaptic inhibition [17,20], stereausis [19,26], and intrinsic MSO properties [11,27] have been suggested to make a major contribution. Despite this abundance of possible mechanisms, it is at present unclear why some neurons follow the slope coding and others the peak coding scheme.

Computational studies have shown that model neurons can achieve sensitivity to sub-millisecond timing differences by a spike timing-dependent plasticity (STDP) rule [28]. By virtue of this learning rule, temporally coherent synapses strengthen when they drive the postsynaptic neuron to spike. STDP can explain the formation of a topological representation of ITDs in the avian brainstem [29–31]. This ITD map derives from a systematic variation in the contralateral latency [29,32–35]. Although the addition of a contralateral delay [36] or well-timed inhibition that precedes the contralateral EPSP [31] can recreate a tuning toward contralateral sound sources, these mechanisms do not explain why neurons with a bITD outside the ecological range are generally tuned to low frequencies, whereas neurons that are tuned to high frequencies typically also show a preference for contralateral sounds, but with a bITD within the ecological range [14–19,37]. Furthermore, persuasive evidence for a mammalian ITD map is lacking [8,11,15,38,39]. This raises the question how MSO neurons obtain their ability to encode sound source directions without the presence of an underlying latency map. Here, we investigate whether this preference of MSO neurons to contralateral sound sources may originate from developmental processes.

## Results

### Juvenile MSO neurons receive binaural excitatory inputs

We have previously used the characteristic field potentials that reverse polarity at the somatic layer of the MSO [10,40] to direct recordings from this layer. To make single-unit or juxtacellular recordings from juvenile MSO neurons, we therefore first confirmed that these field potentials were already present in juvenile gerbils of the age of 15 to 28 postnatal days (P15-28;

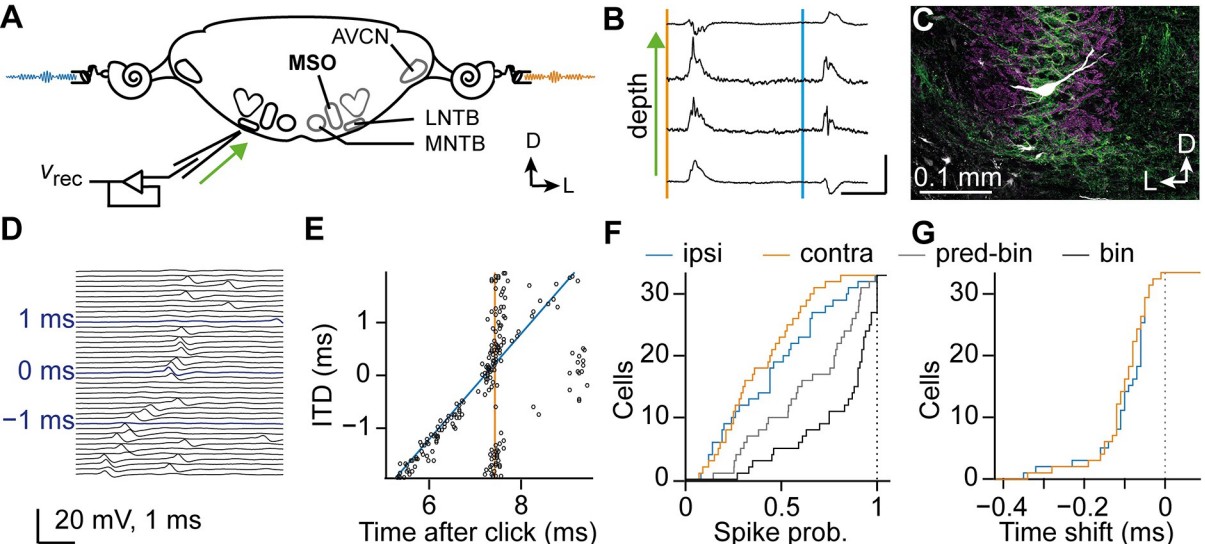

**Fig 1. *In vivo* recordings from juvenile MSO neurons.** (A) Schematic drawing of the auditory brainstem. The MSO is located in the ventral brainstem close to the midline. MSO neurons receive excitatory input from the spherical bushy cells located in the AVCN from both sides, as well as inhibitory input from the ipsilateral medial and lateral nucleus of the trapezoid body (MNTB and LNTB, resp.). The recording pipette approached the MSO from the ventrolateral side and moves mediodorsally with depth (green arrow). (B) Avg. field potentials evoked by clicks at either ear at different penetration depths. Onset of the contralateral and ipsilateral click is indicated by the orange and blue line, respectively. Notice how the ipsilateral field potential reverses from negative to positive more superficially than the depth at which the contralateral field potential reverses from positive to negative [10]. At the place where both field potentials are positive-going, spiking neurons are typically encountered. Scale bars: 1 mV, 10 ms. (C) Biocytin labeling (white) of a principal neuron in the somatic layer of the MSO. Excitatory synapses are labeled by VGluT1 (magenta) and inhibitory synapses by GlyT2 (green). The labeling confirms that the reversal of the field potentials occurred at the somatic layer of the MSO in juvenile gerbils. (D) Clicks were presented at both ears at different ITDs. Single-trial responses of a P15 MSO neuron are shown for ITDs from −2 (bottom) to +2 ms (top). (E) Spike-raster plot of the same neuron as in D for multiple presentations where time is indicated relative to the onset of the contralateral click. Median latency to monaural clicks is shown as a solid line (blue: ipsilateral, orange: contralateral). (F) Facilitated spiking probability for juvenile MSO neurons during binaural stimulation. Cumulative distribution of spike probabilities to ipsilateral ("ipsi"), contralateral ("contra"), and binaural ("bin") click stimuli. Predicted binaural spike probability ("pred-bin") was calculated as $1-(1-P_{ipsi})(1-P_{contra})$ for each cell. (G) Cumulative distribution of time shift of binaural response compared to monaural responses, showing decreased latencies for the first spike response during binaural stimulation for juvenile MSO neurons. The data underlying this figure is available at https://doi.org/10.5281/zenodo.10729468. AVCN, anteroventral cochlear nucleus; ITD, interaural time difference; LNTB, lateral nucleus of the trapezoid body; MNTB, medial nucleus of the trapezoid body; MSO, medial superior olive.

Fig 1B). We confirmed using biocytin labeling that recordings obtained using these field potentials were from MSO neurons located within the somatic layer (Fig 1C).

As the principal brainstem circuit for sound localization is already present at hearing onset [6,41,42], we investigated whether juvenile MSO neurons prefer binaural stimulation. Clicks presented to either ear triggered spikes in juvenile MSO neurons (*n* = 64 cells) as in adult MSO neurons [43]. In a subset of cells, we systematically varied the click ITD (*n* = 33 cells; Fig 1D and 1E) at a click intensity that would not invariably trigger a spike. We observed a facilitation of the spiking probability compared to monaural clicks at contralateral-leading ITDs of 0.25 ± 0.25 ms (avg ± SD; Fig 1F). Probability ratio was 1.4 ± 0.6 (avg ± SD), and only 6 cells had a ratio <1. Moreover, the latency of the first spike decreased during binaural stimulation by 0.10 ± 0.08 ms and 0.10 ± 0.07 ms compared to the ipsi- and contralateral click alone, respectively (Fig 1G; *n* = 33 cells; avg ± SD). This shift likely occurs because the summed EPSP reached AP threshold faster than single monaural EPSPs, indicating the presence of binaural excitatory inputs ("EE"), a characteristic feature of MSO neurons. It furthermore shows that the firing of juvenile MSO neurons is already sensitive to subtle differences in the arrival time of sounds at both ears.

### Best ITDs of juvenile neurons are typically outside the ecological range

The presence of functional inputs from both ears allowed us to investigate ITD tuning of the juvenile MSO neurons by presenting a Zwuis stimulus across a large range of ITDs. Zwuis is a wide-band, multitone complex, which has the property that both the frequencies of the primary tones and of their second-order products are unique [9]. Spiking activity of most juvenile MSO neurons depended on ITD (n = 95 of 106 cells; Fig 2A and 2B). During development, the MSO's dynamic firing range increased (S1A Fig, including 50 cells from adult gerbils, $F_{6,138}$ = 7.5, p = 5.4 $10^{-6}$). Both their dynamic range ($F_{6,138}$ = 12.0, p = 8.6 $10^{-11}$) as well as the maximal slope ($F_{6,138}$ = 18.3, p = 1.5 $10^{-15}$) developmentally increased within the ecological range,

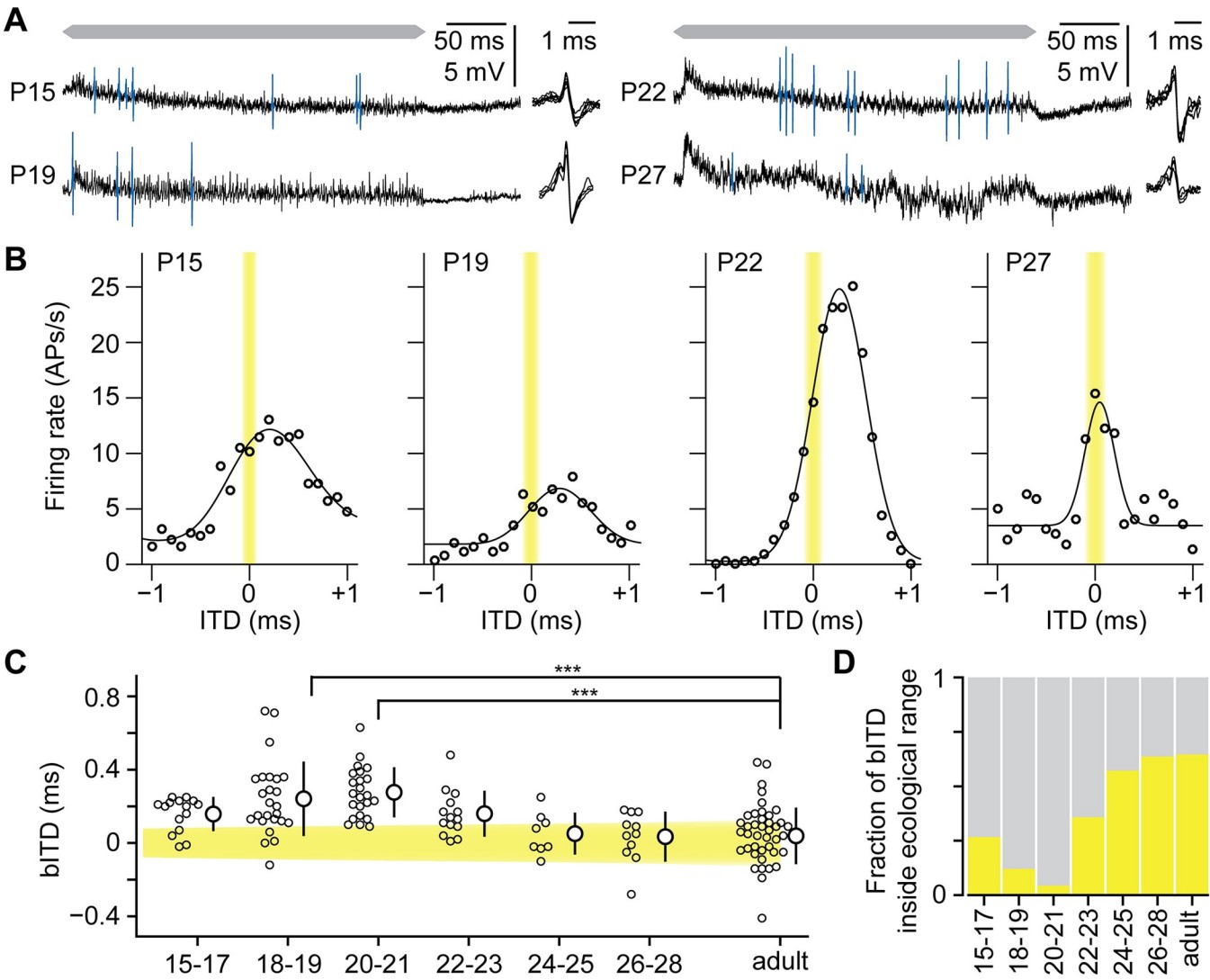

**Fig 2. Best ITDs of MSO neurons shift to the ecological ITD range during development.** (A) Examples of MSO responses at ITD = +0.1 ms at 4 different postnatal ages. APs are colored blue. Example APs are shown at the right of each trace. Gray bar on top indicates period of sound presentation. (B) Average spike rates (open circles) versus ITD for the examples shown in (A). Spike rates were fitted (solid line). Yellow box indicates the ecological ITD range. (C) Developmental changes in the bITD. Age groups are indicated as postnatal days or as adult. Yellow box depicts putative ecological range. Small circles denote individual neurons. Large circles indicate averages with SD. (D) Fraction of bITDs inside the ecological range against age. ***$F_{6,136}$ = 10.02, p = 4 $10^{-9}$, P18-19, Bonferroni-corrected p = 1.7 $10^{-6}$, P20-21, Bonferroni-corrected p = 3.1 $10^{-8}$. The data underlying this figure is available at https://doi.org/10.5281/zenodo.10729468. bITD, best interaural time difference; ITD, interaural time difference; MSO, medial superior olive.

which is ±0.13 ms for adult gerbils [1], indicating that MSO neurons became better at decoding sound source direction. The bITDs of juvenile neurons for the Zwuis stimulus were typically outside the ecological range (Fig 2B and 2C), with the highest deviations observed at postnatal days (P)18-23 (bITDs: P18-19, 0.24 ± 0.20 ms, P20-21, 0.28 ± 0.14 ms, P22-23, 0.16 ± 0.13 ms, avg ± SD). The minimum response was also often outside the ecological range of ITDs. In the second week after hearing onset (P20-27), the bITDs shifted toward the ecological range and became similar to adult bITDs around P26-28 (Fig 2D; P26-28 versus adult: 0.03 ± 0.14 ms versus 0.04 ± 0.15 ms, avg ± SD, and 64% versus 65% within ecological range [10,19]). We conclude that the contralateral bias in bITDs is larger during the first week of development than in adults.

In mammals, the preference for contralateral ITDs is most prominent in neurons that are tuned to low frequencies [17,23,44]. As this is also observed for MSO neurons in adult gerbils (S1D Fig) [19], we tested whether juvenile neurons already have this frequency-dependent tuning. For 76 (out of 95) juvenile neurons, we observed phase-locking to at least one of the Zwuis tones. The best frequency was defined as the tone frequency with the highest vector strength, a measure for phase-locking [45]. Even though the correlation with the best frequency was not strong ($r = 0.3$), the bITDs were on average larger at the lower frequencies in the juvenile neurons (S1E Fig). We tested whether the developmental shift in bITDs only arises for the neurons tuned to sound frequencies of 1.0 kHz and higher. The bITD of the $\geq$1.0 kHz neurons for P15-21 gerbils was 0.25 ± 0.22 ms versus 0.02 ± 0.12 ms for adult gerbils (avg ± SD, 17 and 34 neurons, $t_{21} = 4.1$, $p = 5.1 \ 10^{-4}$). The bITDs of <1.0 kHz neurons shifted toward the ecological range as well (P15-21 versus adult gerbils: 0.24 ± 0.13 ms versus 0.09 ± 0.21 ms, avg ± SD, 32 versus 14 neurons, $t_{17} = 2.4$, $p = 0.03$), but the shift may be less extensive for neurons tuned to low frequencies. To remove the effect of frequency tuning on the bITD, we fitted the bITD against the best frequency with a quadratic function for the adult gerbils (S1D Fig). For each neuron, we then subtracted the bITD expected from the neuron's frequency tuning (S1E and S1F Fig), and these corrected bITDs still deviated toward contralateral delays in juvenile gerbils (Wilcoxon $T_{76} = 1,416$, $p = 3 \ 10^{-4}$), especially at P18-P21 (P18-P19: 0.19 ± 0.23 ms, P20-21: 0.10 ± 0.10 ms, avg ± SD, S1F Fig). Our data therefore show that juvenile MSO neurons have a predisposition to contralateral sounds, which, especially for neurons tuned to higher frequencies, is refined during the first weeks following hearing onset.

## Longer travel times for contralateral sounds in juvenile gerbils

From the juvenile predisposition to contralateral sounds, it can be predicted that the contralateral delay to a brief stimulus should clearly exceed the ipsilateral. To measure these delays, we analyzed the first-spike latency (FSL) to click stimuli. The FSL decreased at higher click intensities by 0.23 ± 0.26 ms and 0.19 ± 0.24 ms for contra- and ipsilateral stimulation, respectively (S2 Fig). For both ears, the monaural FSL was then calculated as the median FSL across all intensities. The ipsi- and contralateral FSL were strongly correlated (Fig 3A, $r = 0.99$), which is partly explained by smaller FSL for neurons tuned to higher frequencies (S2D and S2E Fig, −1.2 ± 0.2 ms/kHz and −1.1 ± 0.2 ms/kHz, $r = 0.57$ and 0.59 for contra- and ipsilateral FSL, respectively) [8]. This correlation between ipsi- and contralateral FSLs also held within animals for which we recorded ≥3 neurons (S2G Fig, $r = 0.91$). The monaural FSLs decreased during development (Fig 3B; ipsilateral FSL, $F_{6,72} = 17.3$, $p = 2.9 \ 10^{-12}$, contralateral FSL, $F_{6,72} = 20.2$, $p = 1.1 \ 10^{-13}$). Although these findings show that ipsi- and contralateral FSLs of individual juvenile MSO neurons are matched, the contralateral FSL was generally larger than the ipsilateral FSL in juvenile gerbils. The difference (ΔFSL) was quite variable at P15-21, but subsequently became fine-tuned toward the ecological range (Fig 3C; $F_{6,72} = 8.5$, $p = 5.9 \ 10^{-7}$). For

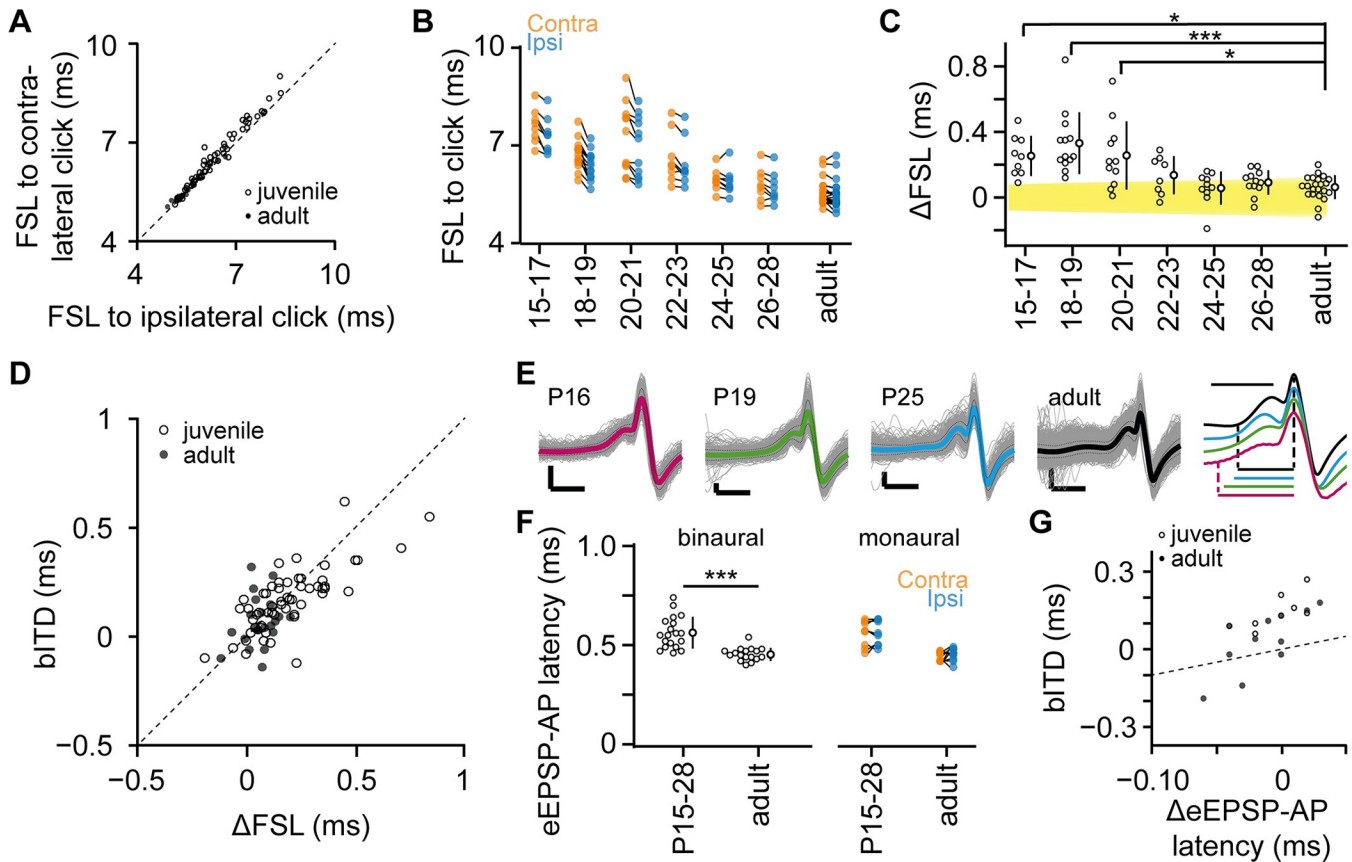

**Fig 3. Monaural travel latencies predict bITDs of juvenile MSO neurons.** (A) FSL to ipsi- and contralateral clicks. Dashed line indicates identity line. (B) Relation between FSL and gerbil age (in postnatal days or adult). Contralateral and ipsilateral FSLs are colored orange and blue, respectively, and latencies from the same cell are connected by a line. (C) Developmental changes in the difference in first spike latency ($\Delta$FSL = contralateral FSL–ipsilateral FSL). Age groups are indicated in postnatal days or as adult. Yellow box depicts ecological range for ITDs. Smaller circles correspond to individual neurons. Averages and SD are shown. $F_{6,72}$ = 8.5, $p$ = 5.9 $10^{-7}$, * P15-17, $p$ = 0.0036, *** P18-19, $p$ = 1.5 $10^{-6}$, * P20-21, $p$ = 0.0012. (D) Relation between difference in FSL and bITD. Value pairs come from the same MSO cell. Dashed line indicates identity line. Pearson's $r$ = 0.7. (E) From left to right, a P16, P19, P25, an adult example and the overlay of the extracellular potential preceding the eAP aligned on the eAP peak. Individual traces are shown in gray. The trace with the median eEPSP-AP is overlaid in color. Scale bars: 0.5 mV, 0.5 ms. The median eEPSP-AP were normalized to AP peak-to-peak amplitude for comparison and are shown with a vertical offset for visual clarity. The eEPSP-AP latency is indicated by a horizontal line below the examples. Scale bar (above the examples): 0.5 ms. (F) eEPSP-AP latency against age groups (left) and monaural eEPSP-AP latency against age groups (right). Value pairs from the same cell are connected by a line. ***Welch's $t_{24}$ = 5.5, $p$ = 1.1 $10^{-5}$. (G) Best ITD (bITD) against the difference in eEPSP-AP latencies evoked by contra- and ipsilateral stimulation ($r$ = 0.7). MSO neurons from juvenile and adult animals are shown as open and closed circles, respectively. Dashed line indicates identity line. The data underlying this figure is available at https://doi.org/10.5281/zenodo.10729468. bITD, best interaural time difference; FSL, first-spike latency; ITD, interaural time difference; MSO, medial superior olive.

71 neurons for which we recorded both the monaural FSLs and the bITD ($n$ = 50 and 21 cells from juvenile and adult gerbils, respectively), $\Delta$FSL was correlated with the bITD (Fig 3D, $r$ = 0.71, bITD = $\beta$ $\Delta$FSL, $\beta$ = 0.69 ± 0.05). This indicates that the longer delay for the contralateral sound-evoked activity underlies the juvenile preference for contralateral-leading ITDs.

## Upstream latencies dictate bITD tuning of juvenile MSO neurons

The SD of the FSL within an animal was 0.3 ± 0.2 ms for both ipsi- and contralateral FSL. If $\Delta$FSL of an MSO neuron would derive from a random combination of FSLs from the same animal, then the SD of its $\Delta$FSL would equal 0.43 ± 0.25 ms (square root of the sum of the ipsilateral and contralateral variances in FSL within a gerbil, $n$ = 8 gerbils with ≥3 recorded MSO neurons). As $\Delta$FSL was less variable (SD: 0.12 ± 0.10 ms, paired $t_7$ = 4.0, $p$ = 4.9 $10^{-3}$),

mechanisms apparently exist that fine-tune the latencies at the MSO neuron. These mechanisms may not only include a selection of inputs that have matching latencies, but also a compensation of any mismatch in latencies at the level of the MSO neuron. With a predominantly somatic origin of the axon, the only latency that the MSO neuron could possibly adjust is the delay that occurs with dendritic propagation [7,27,46]. The EPSP onset at the dendrite is nearly simultaneous with the onset at the soma, but the EPSP peak is delayed by dendritic propagation [7,42,46,47]. The latency from EPSP onset to the AP then includes the dendritic latency together with the spike triggering latency. We therefore analyzed the extracellular EPSP-AP (eEPSP-AP) latencies to investigate whether dendritic latencies had been adjusted. Only for recordings with a seal resistance >20 MΩ, the quality was sufficient to analyze the adult eEPSPs [10], yielding 29 juvenile and 19 adult cells. However, in 11 recordings the maximal rise time of the eEPSP was <1 V/s, which also precluded the quantification of the eEPSP-AP delay. For the remaining 19 juvenile and 18 adult cells, we observed that the eEPSP-AP latency was 0.11 ms longer in juvenile gerbils (Fig 3E, juvenile versus adult MSO, 0.56 ± 0.08 ms versus 0.45 ± 0.03 ms, avg ± SD, Welch's $t_{24} = 5.5$, $p = 1 \times 10^{-5}$). The developmental speedup of EPSPs [6,48] decreases the EPSP-AP latency. However, this decrease in eEPSP-AP latency explained only a minor part (approximately 6%) of the FSL maturation.

If the MSO neuron would adjust the dendritic latencies to compensate for ΔFSL, then the EPSP-AP latency should be smaller for the side with the highest FSL. To compare the dendritic latencies within an MSO neuron, we presented the Zwuis stimulus monaurally to each ear in 19 cells (8 and 11 cells in juvenile and adult gerbils, respectively). Ipsi- and contralateral monaural eEPSP-AP latencies were strongly correlated within MSO neurons ($r = 0.94$, Fig 3F), and did not show a contralateral excess (−0.00 ± 0.02 ms), similar to adults [10]. The small difference in monaural eEPSP-AP latencies was positively correlated with the bITD of the neuron ($r = 0.73$, bITD = α + β ΔeEPSP-AP, α = 0.11 ± 0.02 ms, β = 3.3 ± 0.8, Fig 3G), which argues against dendritic compensation of ΔFSL. While these findings confirm that dendritic latencies contribute to the bITD [27,46], the variation in ΔeEPSP-AP explains only 30% (= 100%/β) of the variation in bITDs. We therefore conclude that upstream delays dominate ITD tuning.

## STDP can account for the developmental shift of bITDs towards the ecological range

The possibility that developmental processes contribute to adult ITD tuning has been suggested by computer simulations that demonstrated that STDP, in which inputs that induce postsynaptic spiking are strengthened, can select inputs with matching delays [28,30]. We sought to explore if such a mechanism could explain the shift in bITDs in light of the observed initial bias in travel times. We designed a computational model in which MSO neurons were given ipsi- and contralateral inputs with a latency difference that corresponded to our observations. These inputs were activated by idealized clicks, resulting in a single presynaptic spike of which the timing was determined by the input's latency, which included an ITD. The neurons were trained with ecological ITDs that resulted in ITD-dependent synaptic activation and postsynaptic firing probability (Fig 4A and 4B). Synaptic efficacy was potentiated or depressed (Fig 4C) when the input was active shortly before or after postsynaptic firing, respectively [28]. This mechanism gradually tuned the bITDs toward the ecological range (Fig 4D–4F) with a retuning rate that depends on the neuron's best frequency (Fig 4D). If the neurons were instead trained with unnatural ITDs, their bITDs shifted to the unnatural ITDs (Fig 4E and 4F), indicating that the STDP mechanism fine tunes the bITDs to the actual ITDs.

Our model includes an STDP rule for which the timing difference for the induction of maximal potentiation versus depression was only approximately 1.8 ms. In contrast, the fusiform

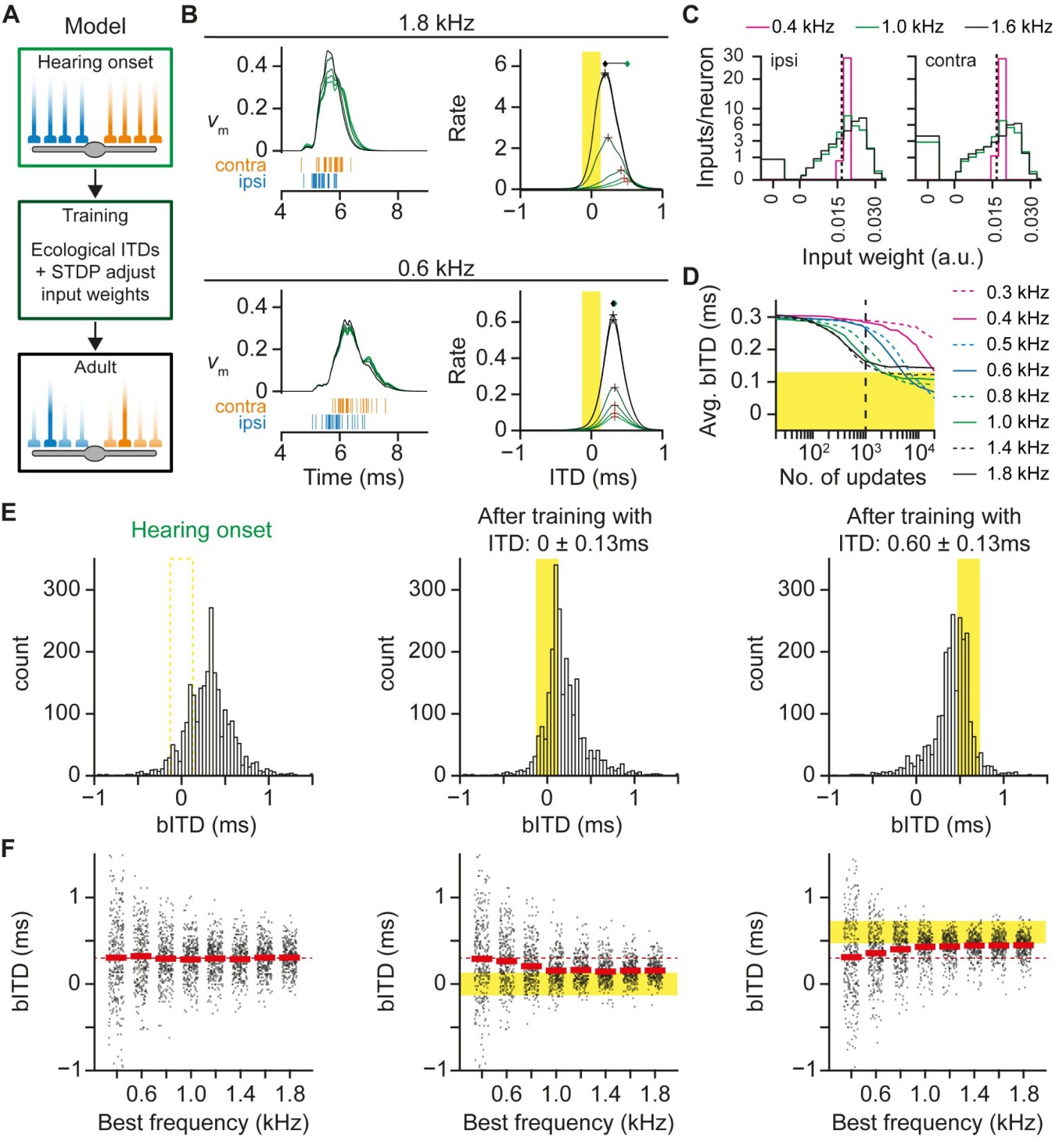

**Fig 4. Spike timing-dependent plasticity can gradually and partially compensate for a latency bias.** (A) The model neuron receives 2 populations of inputs with different latencies. The input latency is determined by a frequency-dependent latency and a side-specific latency. Each input starts with the same synaptic weight. The synaptic weights are adjusted based on synaptic activation relative to postsynaptic firing. The activation pattern depends on ITDs. After a number of training rounds, synaptic weights are different from the starting values, leading to a change in the ITD tuning response of the model neuron. (B) Development of bITDs of model neurons with a best frequency of 1.8 or 0.6 kHz. For each best frequency, the left panel shows the simulated membrane potential (normalized scaling) at ITD = 0 ms (green-to-black traces) and the input latencies (bottom, raster, orange = contralateral, blue = ipsilateral). During the training process, the synaptic weights are adjusted, and therefore the membrane potential changes (green: start, black: learning outcome). The right panel shows the ITD-rate curve during the learning process (from green to black: after 1, 200, 400, 600, 800, and 1,000 updates). Best ITDs are indicated by a plus. On

the top of the graph, the initial (green) versus outcome bITDs (black) are indicated. Each example was trained with ITDs of 0 ± 0.13 ms, indicated by the yellow area. (C) Weights of ipsilateral (left) and contralateral (right) inputs after the learning process for model neurons with a best frequency of 1.6, 1.0 or 0.4 kHz, and 500 neurons per best frequency were trained for 1,000 updates. Vertical dotted line indicates the weight at the start, which was equal for all inputs. Zero weights (or eliminated inputs) are shown separately as a bar. (D) Average bITD against the number of updates for different best frequencies. Neurons with lower best frequencies need more updates to adjust their bITDs to the training ITDs (0 ± 0.13 ms, yellow area). We simulated 500 neurons for each frequency group. We used 1,000 updates (indicated by dashed line) for the other simulations. (E) Histogram of bITDs before (left) and after training with ITDs in the range of 0 ± 0.13 ms (middle; yellow area), similar to the ecological range of adult gerbils, and after training with ITDs in the range of 0.6 ± 0.13 ms (right). With ITDs of 0.6 ± 0.13 ms contralateral instead of ipsilateral inputs are leading by 0.3 ms. Total number of neurons is 2,400 neurons per graph. (F) Best ITD against best frequency before (left) and after training with ITDs in the range of 0 ± 0.13 ms (middle) or ITDs in the range of 0.6 ± 0.13 ms (right). The mean bITD is indicated by the red, solid lines. The training ITDs are indicated by the yellow area. The initial difference in latencies is indicated by the red, dashed line. The code to generate this figure is available at https://doi.org/10.5281/zenodo.10729468. bITD, best interaural time difference; ITD, interaural time difference.

cells in the dorsal cochlear nucleus show STDP for which a timing difference in 10 milliseconds changed the outcome of STDP from potentiation to depression [49], which is an approximately 5 times larger STDP window. We thus asked whether broader plasticity rules can generate the shift in bITDs. We found that as we broadened the plasticity window, its alignment to the spike was critical. In particular, broad STDP rules could generate a shift in bITD toward the ecological range if and only if synapses that were active before the AP were potentiated, while synapses that were active after the AP were depressed (S3A and S3B Fig). Whether this depression resulted from the STDP rule or from a homeostatic mechanism did not matter (S3C Fig), except that the homeostatic mechanism appeared less efficient than STDP-related depression. This homeostatic mechanism could not replace the STDP-related potentiation as it continued until all synapses were depressed, except for one ipsilateral synapse (S3D Fig), eliminating the ITD tuning of the model neurons entirely. STDP can therefore explain the shift in bITD as a potentiation of well-timed, co-activated synapses.

## Discussion

We recorded from juvenile MSO neurons of gerbils throughout their developmental period after hearing onset to study how they acquire the ability to measure ITDs. We found that shortly after hearing onset, MSO neurons can already measure differences in the arrival time of sounds at both ears. Their bITD was typically outside the ecological range, and responses to monaural clicks indicated that this was due to the longer travel time for contralateral signals. Even though their head size increases by >50% during this period, meaning that the gerbils face a constantly increasing range of interaural time cues and that differences between ipsi- and contralateral travel lengths are expected to increase [50], the bITDs showed a developmental shift toward the ecological range. Results of a computer model indicated that STDP might underlie this shift. This mechanism entails that juvenile, slope-coding neurons may develop into adult, peak-coding neurons.

Our conclusions regarding the development of ITD tuning in the gerbil MSO are opposite to those of a developmental study in the DNLL [51]. DNLL neurons receive a mixture of inputs from other brainstem nuclei, including the MSO. The DNLL study compared ITD tuning in P15 and adult gerbils in a subset of cells whose ITD sensitivity resembled those of MSO neurons. On average the bITD was larger in the adult, which they interpreted as evidence that well-timed inhibition shifted the bITD outside the ecological range. In the MSO, local application of strychnine, a blocker of glycine receptors, resulted in a shift of the bITD towards 0 μs, suggesting a role for well-timed inhibition in setting bITD [17,20]. However, the effect of strychnine most likely was an off-target effect on $I_h$ channels [11]. Additional arguments against a role of well-timed inhibition in ITD tuning have been presented [9–11,52,53], and we therefore consider it unlikely to have a clear role in setting bITD during MSO development.

Pharmacological studies would be needed to more conclusively delineate the role of synaptic inhibition during MSO development [10,11,17,20,52,54,55].

Despite the strong difference in interpretation, we found the published ITD tuning at P15 in the DNLL [51] not to be significantly different from our measurements at P15-17 (best interaural phase difference 0.082 ± 0.074 cycles, $n = 9$ in DNLL versus 0.067 ± 0.071 cycles, $n = 10$ cells in MSO; $p = 0.8$). We note that in the MSO, the results at P18-21 showed a stronger contralateral bias than at P15-17. These changes may relate to the relative impact of growth of the brainstem and changes of axonal conduction velocity after hearing onset. In the adult situation, a clear contralateral bias for the bITD was found, but for cells with a frequency tuning >0.6 kHz, which is the large majority of cells in our dataset, more than half had a bITD within the ecological range [51,56], which is also in general agreement with our results. We conclude that our results on the bITD during development are generally in line with previous results obtained in the DNLL.

Apart from STDP and well-timed inhibition, there are other mechanisms that should be considered to underlie the observed developmental shift in bITD. In chickens, a larger conduction velocity in the contralateral branch of the afferent axon compensates for the considerably longer travel distances for contralateral signals [34,35]. This compensation allows the contralateral inputs to function as a delay line [32], as envisaged in the Jeffress model [13]. Some indirect evidence for differential conduction velocities has also been obtained in the cat, as axonal delays estimated from anatomical reconstructions did not match the distribution of bITDs [38]. At the time when functional binaural inputs start to emerge in chickens [57], the contralateral inputs have already begun to speed up, most likely due to myelination [58]. Like gerbils, development in owls is protracted, with for example, myelination taking until the first month posthatch to complete, in agreement with the late maturation of ITD tuning [59]. Although a differential speedup of latencies due to myelination could contribute to a decrease in bITD, it cannot explain the close match between the ipsi- and contralateral latency that we observed for single neurons. It will nevertheless be informative to study in gerbils how myelination and other processes within the auditory brainstem allow the ITD tuning to mature and reach its remarkable precision.

There is currently no physiological evidence for STDP of MSO excitatory synapses, but our simulations together with previous modeling work [28] indicated that STDP presents an attractive alternative explanation for the developmental matching of the bITDs to the ecological range. We observed that STDP could shift the bITD while having a time course that generally fitted STDP observed experimentally for the auditory system [49,60,61]. Three effects observed in our data are predicted by our model. First, the latencies of MSO firing are predicted to decrease as a consequence of STDP. Second, the ipsilateral and contralateral latencies of the synaptic inputs at an MSO neuron are predicted to become similar and their optimal difference will match with the ecological range. Third, the model predicted that the bITD retuning rate is less efficient for neurons tuned to lower tone frequencies. This plasticity mechanism thereby mitigates the initial latency difference by selecting input latencies that match to acoustic ITDs as well as possible, thereby generating "peak-coding" neurons that are fine-tuned to a specific sound source direction.

STDP typically relies on the detection of both a postsynaptic AP and presynaptic activation at a synapse [60]. These mechanisms often involve a rise in calcium concentration which, in order to match synapses by their latencies, needs to be local and synapse specific. Calcium signaling switches from a dendrite-wide to a local transient in developing MSO neurons after P16-20 [62]. The calcium-buffering protein parvalbumin, which is expressed following hearing onset, may contribute to keeping the intracellular calcium transients short [62–64]. Local calcium signals may then arise from N-methyl D-aspartate (NMDA) receptors when they bind

glutamate during presynaptic activation while their magnesium block is relieved by the post-synaptic AP. NMDA receptor-dependent calcium influx was shown to be essential for the potentiation of inhibitory synapses of MSO neurons [65]. Although both NMDA receptor expression and the amplitude of the backpropagating AP decrease after hearing onset [6,62,65–67], calcium influx through NMDA receptors can still elicit potentiation at P30 under certain conditions [65], indicating that NMDA receptors can play a vital role throughout this developmental period. Synapse-specific calcium influx is also generated by calcium-permeable α-amino-3-hydroxy-5-methyl-isoxazolepropionic acid (AMPA) receptors, which become the main AMPA receptor type in MSO synapses after hearing onset [62,68]. Whether the synaptic calcium transient differs when it is paired with a spike, is still an open question [62,65,69–73]. The lower firing rates of developing MSO neurons may contribute to minimizing the overlap of STDP windows and limiting induction of opposing plasticity. We conclude that even though experimental evidence for STDP in juvenile MSO cells is still lacking, our simulations show that it is a promising candidate mechanism underlying the developmental shift in bITDs we observed.

It has been a common finding that ITD tuning for low frequencies, especially for small mammals, lies outside the ecological range, whereas at high frequencies, for most cells it lies within [14–19,22,24,51]. Similarly, in the owls' midbrain the best ITD was negatively correlated with the best frequency, and only fell outside the ecological range at the very low frequencies [74,75]. It has been argued that this correlation between ITD and frequency tuning constitutes an optimal scheme for encoding ITDs [76]. We show here that this optimal coding arrangement can follow from the asymmetry in travel times during development in combination with STDP. The predisposition to contralateral sound sources generates a population coding of ITD as envisioned in the hemispheric channels-model in which all neurons are slope-coding [22]. The developmental fine-tuning of neural delays may alter some slope-coding neurons into peak-coding for which their peak will fall stochastically inside the ecological range. If this developmental fine-tuning occurs more effectively for neurons tuned to higher frequencies (see below), then the optimal coding arrangement may arise during development, with a dual-channel coding for the low-frequency, slope-coding neurons and a homogeneous representation of the ecological range by the bITDs of peak-coding, high-frequency neurons [76].

The initial, large difference in ipsi- and contralateral travel times constitutes a suboptimal pool of fibers for synaptic co-activation, as most fiber combinations have travel differences that lie outside the ecological range. For several reasons, a fine-tuning mechanism involving STDP may subsequently put low-frequency neurons at a disadvantage compared with high-frequency neurons. First, the phase-dependent variation in latencies is likely higher for low-frequency fibers, reducing temporal overlap of the EPSPs. Second, acoustic ITDs constitute only a small fraction of the sound period at low frequencies, in particular, for juvenile gerbils with a smaller ecological range due to the shorter interaural distance, reducing its possibility to compensate exactly for the interaural travel differences. Third, the developmental variation in travel times is at best a single cycle for low frequencies, whereas for high frequencies, cycle skipping can increase co-activation of inputs. In our simulations, low-frequency model neurons needed many more training rounds to tune their bITDs (Figs 4D, S3, and S4). Whereas increasing the number of training iterations is unproblematic in the simulations, biological ITD tuning may be restricted in time to a critical period, and in how much a synapse can change its strength. Furthermore, the input latencies decrease during development [77–79], adding instability to the ongoing fine-tuning process. As a consequence, many low-frequency neurons may retain a tuning to contralateral sound sources analogous to the slope coding scheme [14–19,51]. The gerbil MSO thereby becomes a hybrid of both low-frequency slope coding and high-frequency peak coding neurons for sound source direction.

Our findings suggest that, compared with the avian brainstem in which an ITD map develops from a systematic variation in contralateral delay lines [29,33–35], mammals develop their representation of the ecological range differently. Each MSO nucleus starts with a predisposition to contralateral sound sources, meaning that the nuclei are initially tuned to opposite limits of the ecological range. During a period of fine-tuning, during which the bITDs of MSO neurons decrease, the representation by the two nuclei will progressively cover the frontal sound sources. As these changes can be achieved without any coordination between neurons, an ITD map is not necessary. This mitigation of the contralateral predisposition may appear inefficient, but the predisposition both minimizes the overlap in ITD tuning between MSO nuclei and ensures that the full ecological range will be represented by the adult MSO after the developmental refinement. Our data are thus consistent with the notion that the combination of a contralateral predisposition with synaptic fine-tuning after hearing onset allows Mongolian gerbils to develop the remarkable neuronal ability to exploit the minute timing differences in sound arrival at both ears to localize sounds.

## STAR methods

### Ethics statement

All experiments were conducted in accordance with the European Communities Council Directive (86/609/EEC) and were approved by the national ethical committee (Centrale Commissie Dierproeven, 20171124) and the Erasmus MC animal welfare committee.

### Animals

Breeding pairs of Mongolian gerbils (*Meriones unguiculatus*) were purchased from Janvier Labs. They were kept in the animal facilities of the Erasmus MC. The gerbils had *ad libitum* access to water and food and were given additional bedding to promote nest building. After the female appeared pregnant, the cage was checked daily. Postnatal day (P)0 was defined as the day the litter was found.

Juvenile Mongolian gerbils were anesthetized using an intraperitoneal (i.p.) injection with a mix of ketamine/xylazine (80/15 mg/kg, respectively). Additional i.p. injections with a ketamine/xylazine mix (adult: 16/3 mg/kg, juvenile: 10/0.2 mg/kg) were given when the hind paw reflex started to return. Rectal temperature was maintained at 36.5 to 37.5˚C using a homeothermic blanket (Stoelting Co.). The bony external meatus was exposed to obtain an unobstructed opening to the eardrum. A metal plate was glued to an exposed rostrodorsal part of the skull to fix the head. The animal was placed in supine position and intubated. The animal continued to breathe independently. If breathing became disturbed, the animal was euthanized.

To access the ventral brainstem, we adapted the ventrolateral approach for adult Mongolian gerbils [9–11]. The ventral side of both bullae was surgically exposed and punctured to eliminate any middle-ear pressure differences. Taking care not to damage the tympanic annulus, a small craniotomy was made in the mediodorsal wall of the right bulla at the rostrocaudal level of the first cochlear turn. A small slit was made in the dura to gain access to the ventral brainstem. The opening was usually close to and lateral of the MSO. Care was taken to minimize cerebrospinal fluid entering into the bulla by regularly drying the bulla with paper points.

### In vivo electrophysiology

Loose-patch ("juxtacellular") or unit recordings were made from the MSO neurons using thick-walled borosilicate glass pipettes with filament. The pipette had an open tip resistance of

5 to 8 MΩ when filled with NaCl 135, KCl 5.4, CaCl$_2$ 1.8, MgCl$_2$ 1, HEPES 5 mM adjusted to pH 7.2 with 1 M NaOH. On the day of recording, the pipette solution was sometimes supplemented with 1% to 2% biocytin. The recording pipette was lowered into the brain under high positive intra-pipette pressure (>100 mbar). After breaking through the pia, the pressure was lowered to 30 to 80 mbar. Monaural-alternating click stimuli were used to elicit field potentials in the brainstem [10,40]. The pipette was slowly advanced until the ipsilateral field potential would reverse. Pressure was lowered to 0 to 30 mbar, and the pipette was advanced until action potentials appeared in the field potential as well as spontaneous APs. Recordings were performed in current-clamp mode with bridge balance and pipette capacitance compensation. In case of sudden changes in the recording, the recording was terminated. When biocytin was present in the pipette, to confirm the location of the recording, we either injected it locally using positive pressure or we attempted to electroporate the neuron (Fig 1C). Data were acquired using a MultiClamp 700B amplifier (Molecular Devices), low-pass filtered at 10 kHz with a four-pole Bessel filter, and sampled at 111.6 kHz using custom software (courtesy of Dr. Van der Heijden) written in Matlab 7.6.0 (Mathworks).

### Auditory stimuli

Auditory stimuli were generated using custom MATLAB software and realized through a 24-bit DA processor (RX6, Tucker Davis Technologies (TDT), Alachua), a programmable attenuator (PA5, TDT), and an amplifier (SA1, TDT). Sounds were presented to the animal in a close-field configuration with Shure speakers (22 Hz to 17.5 kHz) that were each connected to a metal bar positioned in front of the external meatus. Sounds presented to one ear were at least 40 dB louder than at the other ear, indicating that cross talk between the ears could be neglected. Click intensity was calibrated using a 1/4" condenser microphone (model 7016, Aco Co). Four oscillations were observed in the first millisecond of the click. The maximum of the peak-to-peak amplitudes of these four oscillations was measured using an oscilloscope and converted to sound pressure level based on the sensitivity of the microphone. The timing difference between 2 speakers was <20 μs.

The following types of sound stimulus protocols were used in this study. We used monaural clicks, a 0.1 ms condensation pulse, with an interval of 30 ms between contralateral and ipsilateral, repeated every 200 ms with varying amplitudes (click-intensity protocol, S2A Fig). The number of repetitions ranged from 15 to 30 per amplitude. Following the click-intensity protocol, we performed a click-ITD protocol. The stimulus was a binaural click with varying ITDs every 250 ms. Click amplitude was set by the experimenter to generate MSO firing regularly, while allowing for some failures to check for facilitation in spike probability. The number of repetitions ranged from 12 to 30. To measure the ITD tuning of the MSO neuron, we performed a Zwuis-ITD protocol [19]. As we did not know the frequency tuning of an MSO neuron beforehand, the Zwuis had the advantage that some of the primary Zwuis frequencies often fell within the frequency tuning of the neuron and therefore activated the neuron. It is a more efficient stimulus, which is important since the juvenile recordings tended to be less stable than the adult ones. The wideband multitone Zwuis ranged between 50 and 5,000 Hz with 30 tone frequencies and random initial phases for the binaural Zwuis. The amplitude of each frequency component was 20 to 50 dB SPL. The Zwuis had a 10 ms cosine-squared ramp at its on- and offset. The total duration of the Zwuis was 300 ms, and every 600 ms it was presented with varying ITD over a range of –2.0 to 2.0 ms with a resolution of 0.1 ms. The number of repetitions ranged from 4 to 10. When there was time, a double Zwuis stimulus was presented monaurally for 60 to 90 s [53]. In initial experiments, only Zwuis-ITD and monaural Zwuis protocols were performed. Later experiments included the click protocols.

Stimulus waveforms were recomputed for every cell, but were kept the same for all within-cell recordings. Presentation of the different stimulus conditions was randomized per repetition.

## Data analysis

All data processing and analysis was performed using custom software written in MATLAB.

**Event detection.**   Extracellular action potentials were detected offline based on a manually set threshold criterion for the maximum repolarization rate of individual events [10]. Only cells for which histograms of the maximum of the negative side of the first derivative showed bimodality were accepted for further analysis.

**Best ITD.**   Best ITD was determined from the Zwuis-ITD protocol, as described previously [19]. Positive ITDs correspond to the signal leading at the contralateral ear. The ITD-rate curve was fitted using a modified Gabor function with an additional power parameter [9]. It was tested whether this fit explained a significant amount of variance. When the fit was not significant ($p > 0.001$), meaning there was no ITD-related modulation, the neuron was excluded from further analyses. For the analysis of the within-ecological range measures, we used ITDs ±0.2 ms as we presented the Zwuis with ITD intervals of 0.1 ms. The adult ecological range was taken to be ±0.13 ms [1]. To estimate the ecological range of juvenile Mongolian gerbils, we measured the interaural distance between the dorsal bases of the pinnae (S4 Fig). The ecological range was then the adult ecological range multiplied by the ratio between the interaural distance and the average adult interaural distance. The interaural distance between P15-28 could be well described by a linear function and, together with the adult ecological range, allowed us to estimate the ecological range as a function of the animal's age (limit = 41.5 μs + 2.6 μs day$^{-1}$; $r = 0.9$), which we used to calculate the ecological ranges at P15-17, P18-19, P20-21, P22-23, P24-25, and P26-28 as ±0.083 ms, ±0.090 ms, ±0.095 ms, ±0.100 ms, ±0.106 ms, and ±0.112 ms, respectively.

**Best frequency.**   Best frequency was calculated from the responses to the Zwuis stimulus at different ITDs. The vector strength [45] was calculated across all trials, and tested for significance (Rayleigh's test, $p < 0.0001$). We simplified the complex tuning of MSO neurons [10,19] by defining the best frequency of the MSO neuron as the Zwuis tone component with the highest, significant vector strength. Phase-locking was less often observed for P15-17 (10 of 15 cells) and P18-19 (18 out of 25 cells) compared to P20-28 (48 of 55 cells) and adults (29 of 30 cells, $\chi^2_3 = 10$, $p = 0.01$).

**Click latency.**   eAPs from +0.5 ms to 10.5 ms following the click were selected. The median latency of this selection was calculated. The selected eAPs within less than 1 ms of the median latency were selected, and the median latency of this subset was taken as the FSL to the click. Next, for each click amplitude the median latency was calculated. To quantify the shift in latency at increased sound intensities, we calculated for each neuron the difference between the FSL and the median latency for the click with the smallest amplitude for which we recorded >3 eAPs.

**Click ITD.**   eAPs were selected within in the range of +2.5 to +8.5 ms following the click for ITDs ≥0.5 ms irrespective of the aural order. The median latency was calculated for the selected eAPs, and the subset with a latency falling within the median latency ±1 ms was selected. From this subset, the baseline ipsi- or contralateral monaural latency was calculated as well as their difference. Binaural effects are expected to occur at the ITD that compensates for this difference. We therefore define binaural ITDs for the click-ITD protocol as the stimulus presentations for which the ITD was within 0.13 ms from the difference in monaural latencies. Next, the median latency was calculated for each ITD. The binaural shift in latency was

quantified as the difference of the average latencies within the binaural range minus the monaural latency. Monaural spike probabilities were calculated from all clicks as the number of eAPs divided by the number of clicks. Binaural spike probability was calculated by counting the number of eAPs and dividing by the number of click pairs. The predicted binaural spike probability was calculated from the monaural failure rates as $1-(1-P_{ipsi})(1-P_{contra})$.

**eEPSP analysis.** Only recordings with a seal resistance >20 MΩ were included. For each eAP, a snippet of the recording was obtained from −5 to +4 ms relative to the eAP peak. This snippet was smoothed by a moving average with a window of 50 μs. The median potential $V_{median}$ was calculated for the smoothed snippet. The standard deviation $SD_{Vsmth}$ of the smoothed potential between −3 to −0.5 ms relative to the eAP peak was calculated. The threshold for the eEPSP onset was calculated as either $V_{median}$+ 2 $SD_{Vsmth}$ or $V_{median}$ + 0.1 (eAP peak potential–$V_{median}$), whichever came first. For every eAP, the first crossing with the threshold prior to the eAP peak was taken as the eEPSP onset. The eEPSP maximal rate of rise was defined as the maximum in the first derivative in the window between the eEPSP onset and −0.2 ms prior to the eAP. If the median eEPSP maximum rate of rise was ≤1 V/s, the recording was excluded for eEPSP analysis. The eEPSP onset distribution was often bimodal. The median difference between the eEPSP onset and the eAP peak was taken as the eEPSP-AP latency.

Monaural eEPSP-AP latency was calculated as above, but from a separate set of recordings of the same cells obtained with the monaural presentation of a Zwuis stimulus.

## Computational model for spike timing-dependent plasticity

**Initialization of neurons.** The underlying mechanism that tunes bITDs may involve a spike timing-dependent adjustment of synaptic strength [28]. To simulate this process, we start with defining neurons that receive inputs. MSO neurons typically receive 7 to 70 inputs per dendrite [7,41]. Here, we assume neurons that receive 30 inputs on each side. Input $i$ is defined by a best frequency ($f_i$), a latency ($l_i$), and a synaptic weight ($w_i$). $f_i$ is related to the best frequency of the postsynaptic neuron $j$ ($bfreq_j$) as follows:

$$f_i = bfreq_j(1 + \gamma \mathcal{U}[-0.5, 0.5]).$$

Here, γ indicates the relative range of frequencies and captures the observation that best frequency of the synaptic inputs are similar to the best frequency of the postsynaptic neuron [19]; $\mathcal{U}[-0.5, 0.5]$ indicates a uniform distribution that ranges from −0.5 to 0.5 from which a value is drawn.

$f_i$ contributes to the latency $l_i$, which is calculated as follows:

$$l_i = l_{offset} + \mathcal{N}(0, l_{SD}) + \mathcal{U}[\varphi_{low}, \varphi_{high}]/f_i.$$

The latency is composed of a fixed part ($l_{offset}$) and two sources of variation; the first derives from a normal distribution $\mathcal{N}$ with a mean of 0 ms and a standard deviation of $l_{SD}$, and the second from a uniform distribution from which a phase is selected, which is divided by the input's best frequency. For half of the inputs of each neuron, $l_{offset}$ was 5.3 ms, corresponding to the contralateral inputs; for the other half, $l_{offset}$ was 5 ms, corresponding to the ipsilateral inputs.

The synaptic weights were initialized to be equal for all inputs and the sum of the synaptic weights ($\Sigma w_i$) was 1. Each input gives rise to an alpha-type EPSP ($v_i(t)$) with an amplitude that

was equal to its synaptic weight $w_i$:

$$v_i(t) = w_i \frac{t - l_i}{\tau_{EPSP}} \exp\left(1 - \frac{t - l_i}{\tau_{EPSP}}\right), t \geq l_i.$$

Initial weights of all inputs were the same. We assume linear summation of $v_i(t)$ [10] to obtain the neuron's membrane potential $v_m(t)$:

$$v_m(t) = \sum_{i=1}^{60} v_i(t).$$

To calculate the bITD of the neuron, we calculate the ITD-rate function of the neuron. For each ITD, we shift the onset of the ipsilateral inputs and calculate $v_m(t)$. From $v_m(t)$, we calculate the instantaneous firing rate $R(t)$ using the following excitability function:

$$R(v_m) = R_{max} \frac{\rho(v_m) - \rho(0)}{\rho(\sum w_i) - \rho(0)}, \rho(v) = \frac{1}{1 + \exp(k_E(v_c - v))}$$

Here, $R(t)$ is calculated from the maximal firing rate $R_{max}$ multiplied by a normalization factor that ensures that $R(t)$ is zero or maximal when $v_m$ is zero or maximal, respectively. $\rho(v)$ is a sigmoid function which calculates the relative firing rate from $v_m$. This excitability relation is expansive [10], and we therefore want $v_m \leq v_c$ in most cases. For every ITD, we calculate the expected firing as the temporal summation of $R(t)$ and divide by 3 ms, which is approximately the duration of the period in which most inputs are active (Fig 4B). The ITD at which the firing is maximal is defined as the bITD of the neuron.

Spike timing-dependent plasticity. The following STDP rule is an adaptation of the rule defined by Gerstner and colleagues [28]:

$$\Delta\omega(t) = \begin{cases} 0.3\exp\left(\dfrac{t - t_{off}}{0.5 t_{scaling}}\right), t < t_{off} \\ 0.5\exp\left(\dfrac{t - t_{off}}{0.5 t_{scaling}}\right) - 0.2\exp\left(\dfrac{t - t_{off}}{5 t_{scaling}}\right), t \geq t_{off} \end{cases}$$

Here, $\Delta\omega(t)$ indicates the amount per AP by which synaptic weight of inputs are adjusted. To adapt the STDP rule to larger time window, we use the $t_{scaling}$ which was a factor 1, 2, 3, 6, or 10. The plasticity rule was shifted by $t_{off}$ to align the rules on either the maximal potentiation or on the time point without any change in weights ($\Delta\omega(t) = 0$, giving a $t_{off}$ of $-0.05 - \tau_{EPSP}$ or $-0.05 - \tau_{EPSP} - (t_{scaling} - 1)(5/9)(\ln(5) - \ln(2))$, respectively. We used $R(t)$ instead of calculating the exact time point of APs. We define that plasticity will only be elicited if $R(t) \geq R_{min}$, which we define as $R(0.15\sum w_i)$. We calculate the thresholded $R(t)$, $R_{thr}(t)$, and then calculate $\Delta w(t)$, the amount by which each synaptic weight is adjusted, as follows:

$$\Delta w(t) = R_{thr}(t) * \Delta\omega(t).$$

$\Delta w(t)$ is used to adjust $w_i$ by the value that arises at the time point at which the input was active:

$$w_i = w_i + \Delta w(l_i).$$

The simulation of the development of bITD tuning does not include a sound stimulus, but we assume that all inputs of the same ear activate all together after which the latencies delay

their postsynaptic effect. The development of bITD tuning is then modeled by the following steps:

1. an ITD is drawn from the teacher ITDs, a uniform distribution within the ecological range;

2. the timing of each input is adjusted to include the teacher ITD plus a jitter ($\mathcal{N}(0, t_{SD})$);

3. $v_i$ is calculated for each input and summated to attain $v_m(t)$;

4. $R(t)$ is calculated, thresholded and then

5. $\Delta w(t)$ is calculated.

6. If $\Delta w$ does not have any value above 0, then redo step 1 to 5, or else we continue by updating the weights $w_i$ using $\Delta w(t)$ and the onset of the respective input's EPSP, including any jitter;

7. $w_i \leq 0$ are set to 0;

8. $w_i$ are normalized so that $\Sigma w_i$ remains the same as at the start.

9. Repeat 1 to 8 until $w_i$ have been updated 1,000 times.

This process is repeated for each neuron. Model parameters are summarized in S1 Table.

## Significance testing

Developmental effects were tested by using the following grouping: P15-17, P18-19, P20-21, P22-23, P24-25, P26-28, and adults (reference group) in a one-way ANOVA. Post hoc *t* tests were performed to test if there was a difference in the mean compared with the adult group. *P*-values were corrected following Bonferroni. Welch's *t* tests were performed if only 2 groups were compared for a difference in their means. Regression was done by a least-squared method using polyfit (Matlab). To test if the frequency-bITD relation for juvenile neurons deviated from the adult neurons, we calculated the difference with the predicted bITD and tested using Wilcoxon signed rank test if this difference was symmetrical around 0, using a normal distribution approximation $\mathcal{N}(\mu: 0, \sigma: \sqrt{(n(n+1)(2n+1)/6)})$ and without correction for ties (4 tied data points).

## Supporting information

**S1 Fig. Developmental changes in the ITD-rate curves and the bITD-frequency relation of MSO neurons.** (A) Dynamic range of MSO firing at different ages. (B) Dynamic range within the adult ecological range of ITDs. (C) ITD sensitivity within the adult ecological range, quantified as the maximal slope of the ITD rate-curve (examples shown in Fig 2B). (D) bITD against the best frequency of adult MSO neurons. Data from reference [19] are reproduced here. Data were fitted by a quadratic function, giving bITD = $\beta_2$ bFreq$^2$ − $\beta_1$ bFreq + $\beta_0$, $\beta_2$: 0.098 ms (kHz)$^{-2}$, $\beta_1$: 0.35 ms/kHz, and $\beta_0$: 0.33 ms. Yellow box indicates adult ecological range. The number in the right bottom corner indicates the percentage of data points within the adult ecological range. (E) As D, but for juvenile MSO neurons. The fitted quadratic function in D is shown again for comparison. Yellow box indicates adult ecological range. The number in the right bottom corner again indicates the percentage of data points within the adult ecological range. (F) The difference between bITD and the predicted bITD against the age of the gerbil. The predicted bITD was calculated from the best frequency of the neuron and the quadratic equation in (D). $F_{6,117} = 6.0$, $p = 1.5 \ 10^{-5}$. Student's *T* test to test if means differ from 0: *** P18-19, $t_{17} = 5.1$, $p = 9.6 \ 10^{-5}$, * P20-21, $t_{20} = 3.2$, $p = 0.0041$. The data

underlying this figure is available at https://doi.org/10.5281/zenodo.10729468.
(TIF)

**S2 Fig. FSL of MSO neurons in relation to click intensity, frequency, and within the same animal.** (Aa) Spike raster plot of a P23 MSO neuron and of a P24 MSO neuron for contralateral (orange) and ipsilateral (blue) clicks ordered along the vertical axis by click intensity. Median FSL is indicated as a dotted line. (Ab) Spike probability against click intensity of the same neurons as in (Aa). (Ac) Median FSL against click intensity of the same neurons as in (Aa). (B) Cumulative distribution of 20% spike probability threshold for juvenile MSO neurons. (C) Cumulative distribution of the time shift of the median FSL compared to the median FSL of the lowest click intensity that elicited $\geq$3 eAPs. (D) Ipsilateral and contralateral first-spike latency (FSL) against best frequency of MSO neurons. (E) Ipsilateral and contralateral first-spike latency expressed in phase against best frequency of MSO neurons. Phase of 0 corresponds to +5.25 ms. (F) Contralateral against ipsilateral FSL to click of MSO neurons from gerbils in which we recorded $\geq$3 neurons ($n$ = 23 cells from 6 juvenile animals, 7 cells from 2 adults). Color corresponds to the unique animal. (G) Contralateral against ipsilateral relative FSL. The grand average of either contralateral or ipsilateral FSL of the animal is subtracted from the absolute contralateral or ipsilateral FSL, respectively. Individual animals are grouped by color as in F. Pearson's correlation coefficients ($r$) are shown in the graph. Dotted lines indicate identity lines. The data underlying this figure is available at https://doi.org/10.5281/zenodo.10729468.
(TIF)

**S3 Fig. Frequency-dependent tuning of the bITD for different STDP rules.** (A–D) Left column shows the broadened STDP rules and the right 4 columns show the average bITD against the number of updates of 400 neurons with a frequency tuning indicated on the top. (A) The broadened STDP rules were aligned on the peak of potentiation. (B) The broadened STDP rules were aligned on $\Delta\omega = 0$. (C) The broadened STDP rules aligned on $\Delta\omega = 0$ without the depression phase. (D) The broadened STDP rules aligned on $\Delta\omega = 0$ without the potentiation phase. In all conditions (A–D), a homeostatic mechanism kept the total synaptic weight constant. The code to generate this figure is available at https://doi.org/10.5281/zenodo.10729468.
(TIF)

**S4 Fig. Ecological range of ITDs of juvenile gerbils.** Developmental change in interaural distance (left axis). For the adult gerbil, the grand average (black), the male average (blue), and the female average (green) of the interaural distances are also shown. The ecological range (right axis) was calculated as the adult ecological range multiplied by the interaural distance normalized to the grand average of the interaural distances of the adult gerbils. The juvenile data was fitted by a linear function. The data underlying this figure is available at https://doi.org/10.5281/zenodo.10729468.
(TIF)

**S1 Table. Model parameters.**
(XLSX)

## Acknowledgments

We thank Dr. Marcel van der Heijden for providing the software for sound stimulus generation and analyses; Yarmo Mackenbach for surgical training; Jean Slenter and Sander Kruithof for support with immunolabeling; and Dr. Aaron Wong, Dr. Peter Bremen, and Dr. Devika Narain for helpful discussions.

## Author Contributions

**Conceptualization:** Martijn C. Sierksma, J. Gerard G. Borst.

**Data curation:** Martijn C. Sierksma.

**Formal analysis:** Martijn C. Sierksma.

**Funding acquisition:** J. Gerard G. Borst.

**Investigation:** Martijn C. Sierksma.

**Methodology:** Martijn C. Sierksma, J. Gerard G. Borst.

**Project administration:** J. Gerard G. Borst.

**Resources:** J. Gerard G. Borst.

**Software:** Martijn C. Sierksma.

**Validation:** Martijn C. Sierksma, J. Gerard G. Borst.

**Visualization:** Martijn C. Sierksma, J. Gerard G. Borst.

**Writing – original draft:** Martijn C. Sierksma, J. Gerard G. Borst.

**Writing – review & editing:** Martijn C. Sierksma, J. Gerard G. Borst.

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
