## [Editor Report · Decision Letter 0]

14 Mar 2023

Dear Dr Borst, 

Thank you for submitting your manuscript entitled "Developmental fine-tuning of medial superior olive neurons mitigates their predisposition to contralateral sound sources" for consideration as a Short Reports by PLOS Biology. Also please accept my apologies for our delay in sending you a decision. We had wished to discuss your manuscript with an Academic Editor but had a bit of difficulty finding someone who was available to provide advice last week.

Your manuscript has now been evaluated by the PLOS Biology editorial staff as well as by an academic editor with relevant expertise and I am writing to let you know that we would like to send your submission out for external peer review.

Once your full submission is complete, your paper will undergo a series of checks in preparation for peer review. After your manuscript has passed the checks it will be sent out for review. To provide the metadata for your submission, please Login to Editorial Manager (https://www.editorialmanager.com/pbiology) within two working days, i.e. by Mar 16 2023 11:59PM.

Kind regards,

Luke

Lucas Smith, Ph.D.

Associate Editor

PLOS Biology

lsmith@plos.org

---

## [Decision Letter · Decision Letter 1]

21 Jun 2023

Dear Dr Borst,

Thank you again for your patience while your manuscript "Developmental fine-tuning of medial superior olive neurons mitigates their predisposition to contralateral sound sources" was peer-reviewed at PLOS Biology. Your study has now been evaluated by the PLOS Biology editors, an Academic Editor with relevant expertise, and by several independent reviewers.

In light of the reviews, which you will find at the end of this email, we would like to invite you to revise the work to thoroughly address the reviewers' reports.

As you will see below, the reviewers appreciate the detailed dataset and analyses provided here which help address an important question. However, they have also provided a number of suggestions to strengthen the study and raise important concerns which would need to be addressed before we can consider your manuscript for publication. We think that the concerns regarding frequency should be carefully considered, and ideally more data would be provided to compensate for the fact that in the adult there is only one low frequency unit. We note that Reviewer 3 has highlighted that the study has not sufficiently accounted for the potential role of inhibition in the model. We think this is, indeed, an important point to consider - however after discussion with the Academic Editor, we would be satisfied with textual changes in response to these concerns, including more discussion and deeper engagement with the literature.

Given the extent of revision needed, we cannot make a decision about publication until we have seen the revised manuscript and your response to the reviewers' comments. Your revised manuscript is likely to be sent for further evaluation by all or a subset of the reviewers.

**IMPORTANT - SUBMITTING YOUR REVISION**

*Re-submission Checklist*

*Published Peer Review*

*PLOS Data Policy*

*Blot and Gel Data Policy*

Sincerely,

Luke

Lucas Smith, Ph.D.

Senior Editor

PLOS Biology

lsmith@plos.org

REVIEWS:

Reviewer #1, Jose L Pena (note, reviewer 1 has signed this review): Very good novel description and analysis of response properties of MSO neurons in juvenile Gerbils and developmental changes of these response properties determining response patterns in this brain region in adult animals.

Potentially significant model of spike timing dependent plasticity, mediating developmental changes of tuning properties and relationship between ITD and frequency tuning of MSO neurons in normal adult gerbils.

Strong results indicating investigated dendritic latencies versus upstream delay lines underlying mechanisms of ITD detection and indicating stronger role of delay lines in ITD tuning of MSO neurons.

Good citation of previous publications.

Overall, a very good manuscript and a few issues and suggestions for clarifying this study:

Figures 2 and 3 indicate 'putative' ecological range of ITD in juveniles and adult gerbils, the Methods section indicates that ITD's ecological range was considered shorter in juvenile than adult gerbils based on changes in head diameter, and reported results of proposed STDP model show that when neurons were trained with different ecological ITD ranges, their best ITD changes shifted. However, previous publications supporting these assumed differences in ITD ecological range along development do not appear clearly cited and whether the change in ecological range has a potential effect on developmental changes in response properties and proposed STDP model is not clearly addressed in the manuscript. 

Lines 112-114: Description of Figure 1E should specifically mention whether this is the spike-raster plot of same neuron as in 1D.

Lines 179-180: Reading description of Figure 3B might become easier if colors representing value pairs for SFL of both ipsilateral and contralateral sounds are specified. 

Lines 220-223: Figure 4A is never called and maybe useful to call it in this part of the text, when mentioning the STDP modeling.

Lines 271-273: a relationship between preferred frequency and ITDs was reported in the midbrain, not ITD detection neurons, of owls. This relationship trend is consistent with the observation that best ITDs become larger for lower frequencies. However in owls, best ITDs are not strongly outside the ecological range. Citing this report supports commonalities of ITD coding properties across species, however it may be useful to specify in the sentence on line 273 that "A relation between frequency and ITD tuning was also found in the owls' midbrain (47)." 

Line 639: Correct apparent typo in legend of supplementary figure S1, where this line should be calling panel F, not E.

Reviewer #2, Michael T. Roberts (note, reviewer 2 has signed this review): This study addresses the important question of how the interaural time difference (ITD) tuning of MSO neurons arises in the weeks following hearing onset. Using in vivo recordings in gerbils, the authors show that the best ITDs (bITDs) of MSO neurons move from well outside the ecological range to within the ecological range over the first two weeks following hearing onset. This shift correlates with a developmental decrease in the difference between the response latencies to ipsilateral and contralateral monaural stimuli within individual MSO neurons, suggesting that the travel times for ipsilateral and contralateral signals to reach MSO neurons are a main driver of ITD tuning. These results further suggest that a developmental mechanism exists to select for similar response latencies between ipsilateral and contralateral inputs. Using a computational model, the authors propose that spike-timing dependent plasticity (STDP) could provide such a mechanism. The model has the additional appeal of providing a potential explanation for why low frequency MSO neurons in mature animals are more likely to have bITDs outside the ecological range than high frequency MSO neurons. Overall, the experiments are elegant and compelling and convincingly support the main conclusions of the study. Some concerns about the authors' interpretation of EPSP to AP latencies, a question of whether an STDP mechanisms is biophysically realistic in MSO neurons, and the need for more clarity about the analyses conducted detract from the study. However, if the authors can satisfactorily address these concerns, the results would represent a significant advance on a problem of central importance to sensory science. 

Major concerns

1. I found the arguments in the "EPSP-AP latencies contribute little to bITD tuning" section difficult to follow, and I believe much of my confusion came from trying to understand how EPSP-AP latencies relate to "the delay that occurs with dendritic propagation" (lines 199-200) or "dendritic latencies" (lines 208, 212). Since the experiments used a single recording electrode, I don't see how the authors can make any direct claims about dendritic latencies based on EPSP-AP latencies. I can see where comparing the slopes of the EPSPs might be informative, as the authors' lab did in their 2013 study (Figure 7 of van der Heijden et al.). This, combined with the EPSP-AP latency measurements might be useful, but on their own, the latency measurements are subject to other interpretations that undercut the authors' dendritic latency argument. Perhaps I am just missing the point here, but if that is the case, I encourage the authors to lay out their arguments more directly and clearly. 

2. The STDP model appears to have a lot of explanatory power and, on its surface, provides a compelling solution to the problem of how individual MSO neurons end up being driven by ipsi and contra afferents with similar FSLs. However, I am concerned that there is no evidence that STDP takes place in MSO neurons, and it is not clear if biophysical mechanisms exist that could support the presumably very brief timing rules needed for STDP driven by binaural stimuli with ITDs <0.13 ms. Most STDP timing rules extend over many milliseconds. Even in the DCN, where the timing rules are relatively brief, the window for LTP is still >10 ms (Tzounopoulos et al., 2004). The authors should explicitly acknowledge this caveat, clearly state or show the time window needed for the learning rules in their STDP model (this might be buried in the parameters, but it should be easy for the reader to pull out), and make it clear to the reader that the mechanisms to support STDP with such a brief timing rule are unknown. At the same time, I commend the authors for clearly explaining in the discussion that there are alternatives to STDP that could explain the refinement of binaural latencies.

Minor concerns

1. Results, line 81. Since the Methods section does not come until the end of the paper, it would help the reader here to clearly specify the age range meant by "juvenile" and to indicate that recordings were made using the juxtacellular approach.

2. Figure 2C and 3C and Results, line 399. The yellow band expands with age because the gerbil head size increases with age, but how did the authors decide on the parameters to use for this? Did the authors measure head sizes themselves or did they base these measurements on a previously published report? In addition, no citation is given on line 399 where the authors describe the change in head diameter.

3. Results, line 165: I found the phrase "which relates to a frequency-dependent latency" difficult to follow until reviewing the referenced figure (S2D-E). It would be more reader friendly to directly state what the relationship is.

4. Results, line 167: It is not clear how "within-animal SD" was calculated. Is this a mean of standard deviations across animals?

5. Results, lines 197-198. The reader needs more info here to understand where the calculated SD of 0.4 ms comes from (i.e., what are the SDs or variances of the ipsi and contra FSLs; I did not see this clearly stated anywhere). Please also directly state the SD of the delta FSL that was observed.

6. Results, line 201 and Methods, line 426. Please standardize how you refer to the seal resistance of the juxtacellular recording. It is described as "input resistance" in the results and "bridge resistance" in the Methods. To me, "seal resistance" would be the preferred term since input resistance is generally synonymous with membrane resistance and bridge resistance is generally an indicator of electrode/series resistance.

7. Figure 4 legend, lines 253-254. The authors state, "Neurons with lower best frequencies need more updates to adjust their bITDs to the training ITDs." Did the authors use the same stimulus duration for low and high frequency stimuli? This would be reasonable, but it also means that high frequency stimuli generate more opportunities for EPSP-AP pairing. Please address this point to make it clear to the reader.

8. Methods, line 425. Typo - "a" should presumably be "(".

9. Figure S1 legend, line 633. Typo - even though it starts a sentence, I believe "BITD" should be "bITD".

10. Figure S1 legend, line 639. Typo - "(E)" should be "(F)".

Reviewer #3: In this study, Sierksma & Borst quantified the distribution of the ITD that elicits peak response rates (bITD) from neurons of the MSO in the gerbil at different developmental stages. The MSO is the primary nucleus for the neural computation of ITD, i.e., the time difference in the arrival of a sound at the two ears, which is the primary cue for sound localization of low frequency sounds. It is by now well established that the bITD in mature mammals, including gerbils, depends on the best frequency tuning of the cell: the lower the best frequency, the larger (more contralateral) the bITD of the neurons. Importantly, for best frequencies below approx. 1kHz, bITDs typically lie outside the physiological range (the max. ITD that the inter-ear distance can generate). In this study, the authors asked how / to what extent this contralateral bITD preference develops during the maturation of the animal. They report that a strong contralateral bias of

---

## [Decision Letter · Decision Letter 2]

29 Feb 2024

Dear Dr Borst,

Thank you for your patience while we considered your revised manuscript "Developmental fine-tuning of medial superior olive neurons mitigates their predisposition to contralateral sound sources" for publication as a Short Report at PLOS Biology. This revised version of your manuscript has been evaluated by the PLOS Biology editors and by one of the original reviewers. Reviewers 1 and 3 were unfortunately not available to re-review this - however in their absence the Academic Editor has assessed the responses to their previous comments. 

Both reviewer 2 and the Academic Editor are fully satisfied by the revision, and we are therefore likely to accept your manuscript for publication based on their assessments. However, before we can editorially accept your study, we need to you address a number of data and policy-related requests, in a revision that we anticipate will not take very long. 

**IMPORTANT: Please address the following editorial requests 

1) ETHICS STATEMENT: Please update the ethics statement in your methods section to specify which institutional animal welfare committee approved this study. Please also add the relevant approval number(s)

2) DATA: You may be aware of the PLOS Data Policy, which requires that all data be made available without restriction: http://journals.plos.org/plosbiology/s/data-availability. For more information, please also see this editorial: http://dx.doi.org/10.1371/journal.pbio.1001797

a. Supplementary files (e.g., excel). Please ensure that all data files are uploaded as 'Supporting Information' and are invariably referred to (in the manuscript, figure legends, and the Description field when uploading your files) using the following format verbatim: S1 Data, S2 Data, etc. Multiple panels of a single or even several figures can be included as multiple sheets in one excel file that is saved using exactly the following convention: S1_Data.xlsx (using an underscore).

b. Deposition in a publicly available repository. Please also provide the accession code or a reviewer link so that we may view your data before publication. 

>>Regardless of the method selected, please ensure that you provide the individual numerical values that underlie the summary data displayed in the following figure panels as they are essential for readers to assess your analysis and to reproduce it:

Fig 1E-G; Fig 2 B-D; Fig 3A-G; Fig 4C,D-F

Fig S1A-F; Fig S2Aa-G; Fig S3A-D; Fig S4

>>Please also ensure that figure legends in your manuscript include information on where the underlying data can be found, and ensure your supplemental data file/s has a legend.

>>Please ensure that your Data Statement in the submission system accurately describes where your data can be found.

3) CODE: Per journal policy, as the code that you have generated is important to support the conclusions of your manuscript, we require that you make it available without restrictions upon publication. Please ensure that any code is sufficiently well documented and reusable, and that your Data Statement in the Editorial Manager submission system accurately describes where your code can be found.

We expect to receive your revised manuscript within two weeks. 

*Published Peer Review History*

*Press*

Sincerely,

Luke

Lucas Smith, Ph.D.

Senior Editor

lsmith@plos.org

PLOS Biology

Reviewer remarks:

Reviewer #2, Michael T. Roberts (Note, reviewer 2 has signed this review): The authors have fully addressed my concerns and should be commended on an excellent study.

---

## [Editor Report · Decision Letter 3]

12 Mar 2024

Dear Dr Borst,

Thank you for the submission of your revised Short Report "Developmental fine-tuning of medial superior olive neurons mitigates their predisposition to contralateral sound sources" for publication in PLOS Biology and thank you for addressing our last editorial requests in this revision. On behalf of my colleagues and the Academic Editor, Jennifer K Bizley, I am pleased to say that we can in principle accept your manuscript for publication, provided you address any remaining formatting and reporting issues. These will be detailed in an email you should receive within 2-3 business days from our colleagues in the journal operations team; no action is required from you until then. Please note that we will not be able to formally accept your manuscript and schedule it for publication until you have completed any requested changes.

PRESS

Sincerely, 

Lucas Smith, Ph.D.

Senior Editor

PLOS Biology

lsmith@plos.org